# Implementation of Virtual Communities of Practice in Healthcare to Improve Capability and Capacity: A 10-Year Scoping Review

**DOI:** 10.3390/ijerph19137994

**Published:** 2022-06-29

**Authors:** Louise Shaw, Dana Jazayeri, Debra Kiegaldie, Meg E. Morris

**Affiliations:** 1Faculty of Health Science, Youth and Community Studies, Holmesglen Institute, 488 South Road, Moorabbin, VIC 3189, Australia; debra.kiegaldie@holmesglen.edu.au; 2Academic and Research Collaborative in Health, SAHHS, La Trobe University, Bundoora, VIC 3086, Australia; dana.jazayeri@unimelb.edu.au (D.J.); m.morris@latrobe.edu.au (M.E.M.); 3Faculty of Medicine, Nursing & Health Sciences, Monash University, Clayton, VIC 3168, Australia; 4Healthscope, Holmesglen Private Hospital, 488 South Road, Moorabbin, VIC 3189, Australia; 5Victorian Rehabilitation Centre, Healthscope, Glen Waverly, VIC 3150, Australia; 6College of Healthcare Sciences, James Cook University, Townsville, QLD 4811, Australia

**Keywords:** implementation, education, health professional, quality, safety, evidence-based practice, digital technology, allied health, health services, nursing

## Abstract

Virtual communities of practice consist of individuals who use a common online platform to share professional expertise and experiences. In healthcare settings a virtual community of practice (VCoP) can optimise knowledge, skills, and the implementation of evidence-based practice. To ensure effective knowledge synthesis and translation into practice, it is essential to clarify the best methods for designing and implementing VCoPs within healthcare organisations. This scoping review aimed to identify the methods used to establish and facilitate online or digitally enabled communities of practice within healthcare organisations across the globe. Six online databases identified papers published from January 2010 to October 2020. Papers were independently screened by two reviewers using Covidence. Data were captured and analysed using a data extraction chart in Covidence. Twenty-four publications that detail methods for establishing a VCoP in healthcare were included. Few studies used a framework to establish a VCoP. It was difficult to identify details regarding methods of development and key elements such as roles, how they were coordinated, and types of technology used. Healthcare organisations can benefit from using a standardised framework for the establishment, implementation and evaluation of VCoPs to improve practice, staff engagement, and knowledge sharing.

## 1. Introduction

Communities of practice (CoP) are networks of individuals who interact regularly to share their interests and develop their knowledge, skills, and capabilities concerning a particular issue [1]. As internet and mobile device use has grown globally, virtual communities of practice (VCoPs) have become more prevalent in healthcare [2,3]. The COVID-19 pandemic limited physical interactions and meetings for sharing of expertise. As a consequence, the relevance and utility of VCoPs became more prominent [4]. Technology based VCoPs can provide opportunities for healthcare professionals to learn, collaborate, and share information without the limitations imposed by geography, cost, organisational boundaries, and time differences [5,6,7]. VCoPs may improve retention of healthcare professionals by reducing perceived professional isolation in rural and regional areas [8]. Health professionals who participate in a VCoP have the opportunity to share ideas, resources and knowledge, as well as access specialised knowledge [9]. VCoPs can also enhance interprofessional collaboration by reducing professional barriers and providing healthcare teams the opportunity to share and implement evidence-based practice [5]. Another benefit is the provision of a risk-free environment for members, increasing the potential for active participation [8]. While VCoPs allow health professionals to stay connected and informed, knowledge sharing in virtual communities can be limited by individual factors (such as active contributions from members), technological factors (such as access to technology and usability issues), and social factors (such as interaction within the group) [9,10,11,12].

Healthcare VCoPs have a variety of forms that allow participants to engage in either synchronous or asynchronous knowledge gain. They typically utilise a range of digital formats to establish a common virtual collaborative space [3]. These include teleconferences, webinars, videoconferences, online meeting spaces, websites, emails, the intranet, and social media [13]. Some VCoPs include blogs, online discussion forums, or file repositories [14]. Project Extension for Community Healthcare Outcomes (ECHO) is an example of a healthcare VCoP [15]. ECHO uses videoconferencing technology that allows primary care providers in rural community clinics to access a multidisciplinary team of specialists [16]. VCoPs can also be established using social media platforms, such as Facebook, Twitter and LinkedIn [8]. For example, Free Open Access Medical education (FOAM) for emergency medicine and critical care [17] communicates via Twitter [18], blogs, and podcasts [19]. Blogs and podcasts that review and present the newest evidence about a topic also allow health professionals to gain easy access to information from clinical experts [20].

The establishment of CoPs in healthcare can be enabled by employing a framework to guide development and ongoing operation [21]. Few studies report how a CoP is designed, implemented, and evaluated in alignment with user needs [3]. Wenger et al. (2002) described seven design principles for developing communities of practice in order to promote development and sustainability [22]. Probst and Borzillo (2008) proposed a CoP governance model based on 57 CoPs from leading business organisations. Their model includes setting clear objectives, sponsorship by senior executives, designation of leadership roles, establishing links beyond the community’s boundaries, ensuring a risk-free environment, and outcome measures to assess the value of CoPs [23]. Barnett et al. (2012) used Probst and Borzillo’s CoP framework to evaluate evidence for VCoPs in medical general practice training [6]. Their proposed health VCoP framework added the extra element of technology with usability, accessibility, and flexibility, of prime importance for knowledge exchange, staff empowerment and learning [6].

VCoPs are becoming increasingly popular in healthcare; however, the best methods to establish VCoPs and which frameworks ensure effective uptake and sustainability have not yet been realised. The characteristics of virtual environments help determine the successful design and management of VCoPs [24]. The aim of this scoping review was to identify the methods used to develop and sustain VCoPs and determine the essential components required to guide the establishment and facilitation of a VCoP for healthcare professionals.

## 2. Materials and Methods

The published protocol paper provides full details regarding the methods used to conduct this scoping review [25]. The review was conducted based on Arksey and O’Malley’s methodological framework [26] and refined using the Joanna Briggs Institute (JBI) methods for scoping reviews [27]. The PRISMA-ScR checklist [28] was followed and reviewed by the research team (LS, DJ, DK and MM) (Appendix A). The PRISMA-ScR checklist has five sections: (a) identifying the research question, (b) identifying relevant studies, (c) identifying the study selection criteria, (d) charting the data, and (e) reporting the results. 

**(a)** 
**Identifying the research question**


The primary research question was (i) What is the extent of reported research on establishing VCoPs in healthcare published in the last 10 years (January 2010 to October 2020)? Additional research questions focused the review to provide guidance for setting up and conducting a VCoP in healthcare: (ii) What methods are used to establish and maintain VCoPs (including frameworks used for VCoP development, who the participants are, how it is coordinated, and methods of communication and knowledge exchange)? and (iii) What potential barriers and facilitators are identified during the implementation of VCoPs? The authorship team included researchers with clinical expertise.

**(b)** 
**Identifying relevant studies**


Eligibility criteria: The population of interest was healthcare professionals who were part of a VCoP for the purposes of building and exchanging knowledge, developing individual capabilities, ensuring their practice is evidence-based, and enhancing interprofessional collaboration. The concept of interest was virtual communities of practice for the purpose of improving clinical outcomes. Communities of practice that described themselves as ‘virtual’, ‘on-line’ or ‘web-based’ were included. The literature included discussed the establishment and maintenance of VCoPs implemented in a healthcare setting for healthcare professionals. The context of interest was any platform used by healthcare professionals to support virtual interactions in healthcare for knowledge advancement and sharing ideas. 

Included articles were accessible as full text and published between January 2010 and October 2020. They were peer-reviewed and written in English. All research methods, such as quantitative, qualitative, mixed methods, systematic reviews, meta-analyses, and guideline implementation, were included. Exclusions included grey literature, commentaries, conference proceedings, and any other opinion pieces. Studies that only evaluated VCoPs, or described VCoPs that were used only for teaching purposes (for example online learning) were also excluded. 

**Search strategy:** The study group followed a three-step search strategy as identified by the JBI [29] in collaboration with an academic librarian, who executed the searches. The final comprehensive search was conducted across PubMed, CINAHL, CENTRAL, PsycINFO, Cochrane Library and Education Resources Information Center. Full details and an example of the search strategy, carried out in PuBMed and CINAHL, are detailed in the protocol paper [25].

**(c)** 
**Study selection criteria**


Studies identified in the search were uploaded to Covidence. The titles and abstracts of retrieved papers were independently screened by two reviewers (LS, DJ). The same two reviewers independently screened full texts to identify studies meeting the review criteria. At each stage, any conflicts were resolved by discussion to consensus, with a third study group member (MM) consulted if necessary.

**(d)** 
**Charting the data**


Data from eligible studies were charted independently by two researchers (LS, DJ) using a data extraction chart developed in Covidence. The tool captured relevant information about key study characteristics identified in the literature, including aims, development and size of the network, stage of development, technological infrastructure, VCoP roles, processes, participants, methods of recruitment, and evaluation. Further detail is provided in the protocol paper [25].

**(e)** 
**Collating, summarising, and reporting the results**


Quantitative and qualitative data from the final dataset were synthesised via an iterative process with new categories and themes identified through ongoing analysis. Thematic analysis was utilised for the qualitative data, and quantitative data were summarised using frequency analysis, with the counts for each category calculated. Further details regarding data synthesis are outlined in the protocol paper [25]. 

## 3. Results

Following removal of duplicates, 2314 records were retrieved from the five databases. The results of the search strategy are shown on the PRISMA flow diagram (Figure 1). Title and abstract reviews identified 2238 records that did not meet the eligibility criteria as detailed in the methods section ‘Identifying relevant studies’. A total of 72 full text articles were assessed for eligibility, and 48 were excluded. The most common reasons for exclusion included: the study was an evaluation study only (*n* = 13); the intervention did not match the research question (*n* = 6); no full text was available (*n* = 5); the setting was wrong (*n* = 5); or the study design was incorrect (*n* = 5). The final review retained 24 articles. 

### 3.1. Study Characteristics

Appendix A outline study characteristics such as year published, study design, and the country in which the study was conducted, which demonstrates the geographic range of included studies. Most studies were conducted in the USA (*n* = 9), Australia (*n* = 4), and Canada (*n* = 3).

### 3.2. Research Methodology

Most of the VCoP studies used a case report design (*n* = 9). Three studies reported using mixed methods [30,31,32]. Other study design descriptions included participatory action research [33] and a discussion paper [34].

### 3.3. Literature Review Papers

Five reviews were identified in the fields of social work [35], GP training [6], health and social care professionals [8], and healthcare [36,37]. These were presented as a rapid review [35], a systematic review [36], integrated reviews [8,37], and a literature review [6]. Broadly, the studies reviewed what is already in the literature regarding VCoPs for certain fields, methods of development, and usefulness. Common themes include the usefulness of VCoPs for increasing knowledge, professional development [8,37], and decreasing isolation [8], as well as the importance of privacy and trust [8,35]. Of note, and unlike other studies, Barnett et al.’s study [6] proposed a new framework for the development of VCoPs based on Probst and Borzillo’s framework, [23] as well as the findings of their review. Their proposed framework included the key themes of facilitation, champion and support, objectives and goals, a broad church, supportive environment, measurement benchmarking and feedback, technology, and community. While further research is needed, this framework may be of use in future studies. Ranmuthugala et al. (2011) sought to understand the reasons for, composition of, and intended purpose of establishing CoPs in healthcare, as well as methods of interaction and outcomes [21]. 

### 3.4. Study Aims

The study aims are summarised in Appendix A. Over half (*n* = 14) of the studies stated that their aim was to report the guidelines or the process for developing and implementing a VCoP, which reflects the inclusion criteria for this study. Many also aimed to evaluate the implications of a VCoP on practice (*n* = 9). Half of the studies (*n* = 12) reported that the purpose of the VCoP was to support the implementation of evidence-based practice and translation of research. Another reported purpose was to conduct a literature review of the evidence supporting VCoPs in healthcare (*n* = 4). Two studies describe the development of an Extension for Community Healthcare Outcomes (ECHO) [16,38].

### 3.5. Background and Purpose of the VCoP

The majority of VCoPs were developed to support sharing professional knowledge, facilitate communication, and foster implementation of evidence-based practice amongst medical professionals and into communities that are geographically dispersed. VCoPs provided the opportunity for health professionals working in rural communities to learn from specialists in more urban areas; supporting doctors in rural practices in South Africa, for example [39]. Other VCoPs included researchers and practitioners with the intent to facilitate the dissemination of research findings into practice [40]. The ECHO system of VCoPs, such as those reported by Lewiecki et al. (2019) [16] and Lewis et al. (2018) [38], provided tele-mentoring through case presentations and didactic presentations, delivering continuing medical education (Appendix A).

### 3.6. Dates

Where stated, almost all VCoPs commenced in the last 20 years, which likely reflects the date ranges imposed on the initial search. Six papers did not state the dates of commencement of VCoPs, or it was unclear [14,30,33,34,39,41]. The longest established VCoP reported was the Cancer Research Network, which has been supported by the National Cancer Institute since 1999 [42]. Project ECHO, the basis for two studies [16,38], was established almost 20 years ago, in 2003 [15].

### 3.7. Methods of Recruitment

There was wide variation in methods used to recruit members. Ten studies that were not literature reviews did not state their method of recruiting participants, or it was unclear [14,16,31,34,38,40,42,43,44,45]. VCoPs often used more than one recruitment method. The most stated method of recruitment was via email (*n* = 4) [32,46,47,48]. One VCoP actively promoted the establishment of the VCoP [47], while in the case of two studies, healthcare professionals volunteered as members after hearing about the study [41,49]. Other methods of recruitment include purposive sampling [39], a hospital survey [33], a face-to-face workshop [30], and recruitment via the researcher [41]. Methods of recruitment are detailed in Table 1. 

### 3.8. VCoP Membership

The types of health professionals most often included in VCoPs were allied health professionals (*n* = 12) and specialist physicians (*n* = 10). Some VCoPs were multidisciplinary; for example, Wolbrink et al. (2017) reported that their VCoP involved a physician/consultant, fellow, resident, medical student, nurse, nursing student, nurse practitioner, and respiratory therapist [44]. Other VCoPS were interdisciplinary to advance treatment in a particular area. For example, the VCoP described by Pratte et al. (2018) consisted of paediatric physiotherapists working with children with developmental coordination disorder [30]. Alary Gauvreau et al.’s (2019) VCoP involved 13 speech-language pathologists working in aphasia rehabilitation [41]. See Table 2.

### 3.9. Sponsorship and Leadership Roles

Three VCoPs [30,41,47] were sponsored by the authors reporting on the VCoP. For example, Alary Gauvreau’s evaluation study for speech-language pathologists in aphasia rehabilitation was undertaken as part of her PhD [41]. Landes et al. (2019) reported on the building and implementation of a VCoP for dialectical behaviour therapy for the Department of Veterans Affairs that she created as a psychologist, researcher, and employee of Veterans Affairs [47]. Other VCoPs were joint ventures between healthcare organisations and universities; for example, a community of practice for dementia champions between Health Education England and the University of Hertfordshire [34], and a social work VCoP was a partnership between the University of Buffalo School of Social Work and the University of Buffalo Teaching and Learning Centre [45]. Other VCoPs were supported by large organisations such as Project ECHO [16,38] and the National Cancer Institute [40]. 

Facilitators or leaders were reported to drive the community and encourage members to participate. They may, for example, coordinate the preparation and conduction of meetings [46]. The Swedish Oral Medicine Network’s monthly meetings were led by a chairperson, but the meeting’s facilitation rotated among core members [43]. For a VCoP formed for the purpose of General Practitioners’ continuing professional development, the facilitation team comprised specialist physicians, senior GPs, a dedicated content facilitator, and an information technology administrator [48]. Additional roles included clinical experts, champions, and knowledge brokers, which are described in Appendix A.

### 3.10. Technology and Infrastructure

A wide variety of web and internet tools were used to facilitate VCoPs; this allowed members to access VCoPs from anywhere, promoting active participation and interaction for health professionals who are isolated or have time limitations. Many VCoPs utilised more than one type of technological infrastructure. Over half (*n* = 14) of the studies reported using a web-based or digital platform (e.g., Web 2.0 or Microsoft SharePointWeb). The remainder used videoconferencing (e.g., Adobe Connect, Zoom) (*n* = 3), social network sites (*n* = 1), and podcasts (*n* = 1). Table 3 details the technological infrastructure of the reviewed VCoPs.

### 3.11. VCoP Development

The extensive variety of presentations of VCoPs revealed considerable variability in how networks were developed (see Appendix A). For some VCoPs, development was planned and organised. For example, Lewis et al. (2010) described the development of a VCoP for the University of Buffalo’s School of Social Work using Wenger’s developmental stages of a CoP [45]. Lewis et al. (2010) outlined how project goals and aims were developed and how these were matched to the VCoP technology platform [45]. TeleECHO networks were developed with the support of a dedicated team from Project ECHO which conducted training for those committed to starting and developing this method of VCoP [15]. Conversely, other VCoPs developed less purposefully, and possibly evolved from pre-existing conference meetings; for example, Friberger (2013) reported that in the early 1990s, four hospital clinics discussed cases via telephone conferencing, which evolved into the Swedish Oral Medicine Network (SOMNet) [43]. 

### 3.12. Size of the Network

The size of the network for the VCoPs was not always reported in the papers. Where it was reported, size varied greatly and was reported in multiple ways, including the number of individual members or participants, hospitals, and research centres, as well as the number and range of countries included in the network. For example, one VCoP included 13 speech and language pathologists across a city [41], compared to an online platform such as OPENPedatrics, which engaged a global community of critical care clinicians [44]. The VCoP model ‘Project ECHO’, which was discussed in two papers [16,38], reported participants in more than 9000 cities and 180 countries since launching [15]. Other papers where the VCoP included a website also reported the number of page views and site visits in a particular time period; for example, Ting (2018) stated that the CanadiEM website (at the time of reporting) had more than 2.5 million page views from 217 countries [49]. The size of the networks is detailed in Appendix A.

### 3.13. VCoP Forms of Interaction

A wide range of formats for VCoP interactions were reported, although not all papers provided detailed descriptions of the activities that were included (Table 4). Several VCoPs (*n* = 8) offered a learning hub platform (e.g., [30,48]) with resources available on a website and asynchronous learning occurring via an online discussion forum, for example. Others offered live discussions (for example, case studies, clinical questions, and hot topics), online meetings, and live presentations by a guest speaker or clinical expert. An online discussion platform via forum posts was the most reported activity (*n* = 9), followed by online live discussions (for example, case-based presentations, clinical questions, and hot topics) (*n* = 7). Other activities included periodic emails with news, updates, or notifications of forum posts (*n* = 5). Some VCoPs reported providing recordings of live meetings (*n* = 3), online-live presentations (*n* = 3), YouTube videos, or webinars (*n* = 2). Less commonly reported activities were blogs, tele-mentoring, and podcasts. 

### 3.14. Theoretical Frameworks Supporting Development of the VCoPs

Six VCoPs did not report using a specific framework [8,14,30,34,43,48]. The most reported framework for the development of VCoPs was Wenger’s CoP theoretical framework (*n* = 4) [31,40,45,49]. Two VCoPs were part of an overarching model, such as Project ECHO. The ECHO model learning framework involved a virtual community of participants who presented anonymized cases to specialists and their peers for discussion and recommendations [15], such as Bone Health TeleECHO [16] and Dermatology ECHO [38]. Probst and Borzillo’s (2008) framework of successful factors for VCoPs [23] formed the basis of one of the reviewed VCoPs, which was a web based VCoP for primary health workers in the Basque Health System [46]. Following a literature review of General Practice training and VCoPs, Barnett et al. (2012) developed a health VCoP framework [6] that was also based on Probst and Borzillo’s business CoP framework [23]. Two reviewed papers reported using Barnett’s health VCoP framework [35,47]. Theoretical frameworks employed for each VCoP are detailed in Appendix A. The number and types of frameworks used are noted in Table 5.

### 3.15. Outcome Measures

Although ‘evaluation only’ studies were excluded in this scoping review, many included studies had an evaluation component. The methods employed for outcome measurement also varied; quantitative outcomes such as level of participant engagement and web analytics being common evaluation tools. Landes et al. (2019), for example, used web analytics to measure usage trends, including the number of unique users each month, user locations, visits, and average number of requests [47]. Qualitative outcomes included member surveys, semi-structured interviews, and focus groups regarding members’ views on enablers, challenges, processes and outcomes. 

Many studies used mixed methods to assess the level of impact of the VCoP. Farrell et al. (2014) claimed that web analytics alone would not capture the full extent of the growth and member engagement of the CoP; in addition to page views and membership data, they collected the level of response to different discussion postings and regular member surveys [40]. Wolbrink et al. (2017) combined web-based member surveys with site analytics to evaluate their online paediatric critical care CoP [44]. SOMNet evaluation methods included an online questionnaire, interviews with participants, and observations of teleconference meetings [43]. Lewis et al. (2018) presented findings from a case study using photographs to demonstrate how the ECHO VCoP worked to improve both provider knowledge and patient outcomes [38]. Outcome measures employed are detailed in Appendix A.

## 4. Discussion

This scoping review showed rapid progress in the development of virtual communities of practice across the globe to improve evidence-based practice and clinical outcomes in healthcare. Several methods were used to develop and maintain VCoPs, such as webinars, online discussions, blogs, live case discussions, and use of social media. There was, however, little consensus regarding the best way to approach VCoP design and implementation in healthcare settings. Extensive variability was observed in the types of participants, how they were coordinated and facilitated, methods of access, communication approaches, and digital support. Dube et al. noted that VCoPs in organisations share some common characteristics, yet they all have ‘unique personalities’ [13].

The evidence synthesis also highlighted wide variations in the quality of reporting in the literature reviewed. Thirteen full text articles were rejected as they only evaluated the VCoP and did not provide details about methodology. The process of establishing, developing, and maintaining VCoPs was also inadequately reported. A lack of description of the key elements of the development of a VCoP makes it difficult for others to replicate interventions and identify best practices for how additional CoPs can be best designed to facilitate sharing information and knowledge [36]. 

Wenger et al. advised that for CoPs, it is important to know the best methods of development and implementation to benefit participants as well as the host organisation [22]. Our findings highlighted that healthcare VCoPs are diverse in nature and not always implemented in a systematic manner. A small number of studies used a framework to design and develop their VCoP. Four reviewed papers [31,40,45,49] utilised Wenger’s principles for the development of a CoP [22], which suggests that these principles could be successfully applied. The additional elements of mode of interaction and technology employed could also be considered. Two studies [35,47] employed Barnett et al.’s framework [6] for the development of their VCoPs. Barnett et al.’s proposed health VCoP framework [6], developed from Probst and Borzillo’s ten commandments for VCoPs gleaned from the business literature [23], added the elements of technology and community [6]. These two elements may ensure ease of use and access, with options for both synchronous and asynchronous communication [6]. Further evaluation is warranted to determine whether this framework could be applied to the development of future international healthcare VCoPs. An action research framework was suggested in one paper, with the development of the CoP recorded by facilitators, in addition to community projects that were devised and implemented [50]. The use of a standardised framework may assist a quality assurance process to ensure all key elements are considered in the design and reporting of the establishment of VCoPs for healthcare. 

Important factors in the growth of VCoPs for healthcare professionals in a 7-year longitudinal study was the presence of a centralised leadership structure and the frequent rotation of leadership over time [9]. Few papers in this scoping review detailed the key organisation or sponsor of the VCoP. A VCoP pre-implementation survey for general practice training revealed that potential members perceived a VCoP’s sponsor to be important as an initial stakeholder champion [2]. Other roles described in the reviewed papers included leaders, facilitators, subject matter experts, and content editors; however, who performed each of these roles was seldom indicated. Sometimes, the researchers reporting on the VCoP, Galheigo et al. (2019) [33] and Alary Gauvreau et al. (2019) [41], for example, also held VCoP roles such as facilitators or leaders.

An extensive range of technologies and formats for VCoP interactions were reported. VCoPs depend on active participation by all members to develop and share content. All members need to be encouraged to contribute to discussions that enable the VCoP to be successful and sustainable [8]. However, methods of interaction within VCoPs are likely to influence members’ experiences [3]. Online live discussion forums were popular, suggesting that their social nature helped to create a sense of community among members [8]. A key factor in VCoPs’ success is members’ perceived trust and commitment to both each other and the organisation, which may be assisted by live discussion [20,42]. Member engagement can also be facilitated by ensuring active facilitation, a stimulating environment [51], and employing technologies that allow for a range of communications and ease of use [8]. Healthcare professionals’ access to digital technologies, their technical capability, and the usability of VCoP platforms also need to be considered when developing a healthcare VCoP [9]. Digital healthcare capability frameworks that could facilitate the implementation of practice into real-world settings, have recently been published [52,53,54,55]. These frameworks highlight that health professionals benefit from an understanding of digital health technologies that enable secure connection, collaboration, and information sharing with other medical and health professionals; the potential benefits of engaging in online networks; and the technologies that support collaborative relationships [52].

This review has several limitations. Only literature published in English was reviewed; there may be other global examples of VCoPs in non-English speaking cultures. We did not analyse the digital technologies used to support VCoP, such as sensors, webcams, or robotics. The lived experience of VCoP users was beyond the scope of this evaluation. We did not evaluate how participation in VCoPs facilitated work-based learning, work satisfaction, staff retention, or career pathways. We also excluded non-empirical studies and reports that were purely qualitative in nature. Finally, the outcomes of VCoPs on common health services problems, such as hospital falls [56], awaits confirmation with controlled trials.

## 5. Conclusions

A variety of methods are available to establish and maintain VCoPs in healthcare. However, as a result of poor reporting on the development and key elements of VCoPs, the overall data quality was limited and difficult to extract for analysis. It was often difficult to determine if VCoPs were CoPs with a technology component. Many identified studies focused on evaluating VCoPs, members’ experiences, and user analytics, rather than detailing the methods and any theoretical frameworks used to establish the VCoP. Methods adopted depend on needs, the personal preference of participants, access to digital technologies, access to multi-media platforms to develop education and training resources within the VCoP ‘toolkits’, the time available, and support from organisational leaders. Healthcare organisations may benefit from using a standardised framework for the establishment, implementation and review of VCoPs to improve clinical practice, staff engagement, and knowledge sharing.

## Figures and Tables

**Figure 1 ijerph-19-07994-f001:**
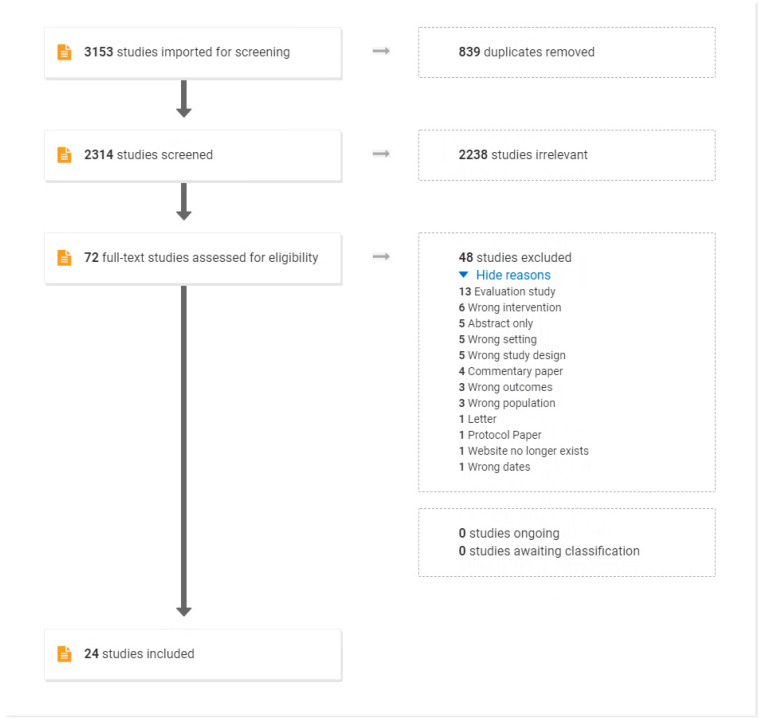
PRISMA Diagram of Scoping Review Results for VCoPs in healthcare.

**Table 1 ijerph-19-07994-t001:** Methods of recruitment for VCoP participants.

Methods of Recruitment for Participants	Number of Reports ^1^
Email	4
Self-volunteere after hearing about study	2
Purposive sampling	1
Hospital survey	1
Recruited by/via researcher	1
Actively promoted (e.g., on relevant websites or email banner)	1
Not stated	10

^1^ More than one type of method of recruitment were, at times, reported in each paper. The table presents the number of reports of each type of activity across the reviewed papers. Not all papers reported the method of recruitment for VCoP participants.

**Table 2 ijerph-19-07994-t002:** Membership population of VCoP.

Population Description	Number of Reports ^1^
Allied health clinicians	12
Specialist physicians	10
Researchers	4
GPs	3
Nurses	3
Medical students	2
Healthcare managers	1
Dentists	1
Not specifically stated (e.g., ‘primary health care professionals’) and ‘health and social care professionals’	4

^1^ More than one type of population were, at times, reported in each paper. The table presents the number of reports of each type of activity across the reviewed papers. Not all papers reported the population. Review articles were also included.

**Table 3 ijerph-19-07994-t003:** Technological infrastructure of VCoPs.

Technological Infrastructure	Number of Reports ^1^
Web based/digital platforms (e.g., Web 2.0, Microsoft SharePointWeb)	14
Videoconferencing (e.g., Adobe Connect, Zoom)	3
Email (e.g., distribution of newsletter)	2
Social network sites	1
Podcasts	1
Platform not stated, but discusses issues such as ease of use, access, flexibility	1

^1^ More than one type of technology were, at times, reported in each paper. The table presents the number of reports of each type of activity across the reviewed papers. Not all papers reported technological infrastructure.

**Table 4 ijerph-19-07994-t004:** VCoP forms of interaction.

Forms of Interaction	Number of Reports ^1^
Online discussion platforms ( e.g., forum posts, case studies, clinical questions)	9
Online live discussions (e.g., case-based, clinical questions, hot topics)	7
Web-based provision of resources	8
Online meetings	5
Emails with news/updates/notifications	5
Provision of recorded meetings	3
On-line live presentations	3
YouTube videos/webinars	2
Blog	1
Implementation registry	1
Tele-mentoring	1
Guest speaker/clinical expert	1
Podcast	1

^1^ More than one type of activity can be reported in each paper. The table presents the number of reports of each type of activity across the reviewed papers.

**Table 5 ijerph-19-07994-t005:** Theoretical framework for development of VCoP.

Theoretical Framework	Number of Papers
No specific framework	6
Wenger’s CoP framework	4
Barnett et al.	2
Probst and Borzillo	2
Project ECHO	2
Theoretical concepts of constructivism, social learning, situated learning	1
Proposed social framework	1
Implementation registry	1
Participatory action research	1
World shared practices	1
Developed cluster map	1
Not applicable	2

## Data Availability

The data presented in this study are available on request from the corresponding author.

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
