# Peer review of "Implementation of Virtual Communities of Practice in Healthcare to Improve Capability and Capacity: A 10-Year Scoping Review"

_ijerph, 2022, doi:10.3390/ijerph19137994_

Round 1

Reviewer 1 Report

The manuscript by Louise Shaw et al reviewed 24 papers and try to identify the methods used to develop and sustain VCoPs and determine the essential components to guide the establishment and facilitation of a VCoP for healthcare professionals. Although the author has summarized the existing literature from multiple perspectives, as a review article, the current manuscript has not yet reached the recommended publication level in IJERPH. Some of the main comments are as follows:

1)      The charts in the full text are too monotonous and simple. It is recommended that the author use a bar chart or other methods to display the results, rather than a single number list. This single summary method does not allow readers to see clear progress in the field.

2)      Although the authors screened and identified 24 studies for review, this number is very limited or insufficient for a review. The authors are advised to substantially increase this number to make the results more convincing.

3)      As a reader, it is very important to get the author's logic and ideas, as well as the thinking based on these methods, however, these things cannot be seen clearly from the current manuscript. The author should add more deep thinking, discussion, and suggestions. These points are an important part of a review and the significance of evaluating existing methods in a related fields.

Author Response

RR Reviewer comment

Response

Line

)      The charts in the full text are too monotonous and simple. It is recommended that the author use a bar chart or other methods to display the results, rather than a single number list. This single summary method does not allow readers to see clear progress in the field.      

This scoping review does not attempt to show progress in VCoPs over time. The aim was to explore the methods used to develop and sustain VCoPs and determine the essential components to guide the development of a VCoP for healthcare professionals. The summary counts allow readers to determine how often different elements were detailed in the reviewed papers. The tables have been re-ordered to allow readers to see what the most commonly reported methods for each key element of the VCoP were.

230, 243, 277, 321

 Although the authors screened and identified 24 studies for review, this number is very limited or insufficient for a review. The authors are advised to substantially increase this number to make the results more convincing.

Scoping reviews are a type of evidence synthesis that have the objective of identifying and mapping relevant evidence that meets pre-determined inclusion criteria regarding the topic under review, in this case, VCoPs. A rigorous search strategy method was followed for the scoping review, which is now internationally recognised best practice for conducting scoping reviews (https://jbi.global/scoping-review-network/about). We followed the three-step approach advised for the conduct of scoping reviews. The full search strategy is detailed in our protocol paper. Six databases were comprehensively searched in collaboration with an academic librarian. I have now included in the paper detail on which databases were searched. After applying our study selection criteria, only 24 studies were included. This suggests more work needs to be done that details the establishment of VCoPs in healthcare.

111 -140

As a reader, it is very important to get the author's logic and ideas, as well as the thinking based on these methods, however, these things cannot be seen clearly from the current manuscript. The author should add more deep thinking, discussion, and suggestions. These points are an important part of a review and the significance of evaluating existing methods in a related field.

Some detail has been added in the conclusions about the difficulties in conducting the scoping review. As described earlier, scoping reviews synthesise the literature on a particular topic, identify key concepts and gaps in the research. We feel that our paper has identified different types of healthcare VCoPs, analysed knowledge gaps in the establishment and maintenance of healthcare VCoPs, clarified key elements and concepts, and made suggestions for the development of healthcare VCoPs, using a framework.

434 - 446

Reviewer 2 Report

The abstract does not convey as effectively as the first paragraph of your paper the idea of what a VCoP is and why studying them is relevant, and this could be polished up. Conversely, your opening paragraphs in the Introduction section are brilliant in summarising your object of study and justifying why it deserves academic attention.

Within this introduction you provide an extensive range of references that help to build a good enough context for your paper -this is notably one of the strengths of your work. The same goes for the theoretical support for your methodology, which is concisely referenced yet strong enough to understand what you have done and how it is sustained by previous methodologies.

As for the methodology itself, I believe the way you dissected visibilises the process and in your case is, yet again, another big strength; since you have planned carefully how to systematically address your research questions and how to approach its investigation in such a rigourous way. That you explain all that so comprehensively in barely three pages is remarkable. I would only advice, as an area with small room for improvement, to explain a bit more detailedly your remark in lines 148-149 on how title and abstract reviews led to the exclusion of 2238 records: you have hinted the inclusion criteria before; but, since this is such a key step in your process of narrowing down the records studied, this passage, more than any other probably, allows some extra explanation.

Your text is also very well-written - I'd just advice a final revision to refine some minor typos (i.e. "As internet and mobile device use have" instead of "As internet and mobile device use has" in l. 39 -I assume that "use" here is premodified by both "Internet" and "mobile device"; but if you present "Internet", on the one hand, and "mobile devide use" on the other, it would work).

I also liked your discussion section, since it clearly connects your research questions with the findings in the results section without being too rigid. Content-wise, the only area that left me a bit cold was the conclusions: too brief for such a notable paper. Part of it, I assume, is that ideas that would traditionally fall in this epigraph have been presented in the Discussion section. However, I think your study could be empowered with some further reflections on the impact of your findings and future lines of research. Some of which, as I said, are hinted, for instance, when discussing your limitations previously: but you can expand these thoughts in the conclusions without changing a single comma of your research, and the final result would look even more solid.

Author Response

Reviewer comment

Response

Line

The abstract does not convey as effectively as the first paragraph of your paper the idea of what a VCoP is and why studying them is relevant, and this could be polished up. Conversely, your opening paragraphs in the Introduction section are brilliant in summarising your object of study and justifying why it deserves academic attention.

Thank you, this has now been amended and the opening sentences read: ‘Virtual Communities of practice consist of individuals who use a common online platform to share professional expertise and experiences. In healthcare settings they can optimise knowledge, skills, and the implementation of evidence-based practice. To ensure effective knowledge synthesis and translation into practice, it is essential to clarify the best methods for designing and implementing VCoPs within healthcare organisations…’

20 - 22

I would only advice, as an area with small room for improvement, to explain in a bit more detail your remark in lines 148-149 on how title and abstract reviews led to the exclusion of 2238 records: you have hinted the inclusion criteria before; but, since this is such a key step in your process of narrowing down the records studied, this passage, more than any other probably, allows some extra explanation.

This is detailed fully in methods section b ‘Identifying relevant studies’. I have made this more explicit in the results.

111 - 134

Your text is also very well-written - I'd just advice a final revision to refine some minor typos (i.e. "As internet and mobile device use have" instead of "As internet and mobile device use has" in l. 39 -I assume that "use" here is pre-modified by both "Internet" and "mobile device"; but if you present "Internet", on the one hand, and "mobile device use" on the other, it would work).

Amended, thank you.

41

Content-wise, the only area that left me a bit cold was the conclusions: too brief for such a notable paper. Part of it, I assume, is that ideas that would traditionally fall in this epigraph have been presented in the Discussion section. However, I think your study could be empowered with some further reflections on the impact of your findings and future lines of research. Some of which, as I said, are hinted, for instance, when discussing your limitations previously: but you can expand these thoughts in the conclusions without changing a single comma of your research, and the final result would look even more solid.

Further detail has been added to the conclusions: ‘Poor reporting on the development and key elements of VCoPs, meant the overall data quality was limited and difficult to extract for analysis. It was often difficult to determine if the VCoPs were CoPs with a technology component. Many of the identified studies focused on evaluation of the VCoP or members’ experiences and user analytics, rather than detailing the methods used to establish the VCoP and theoretical framework.’

434 - 446

Reviewer 3 Report

It is a very interesting work which will be improved  if:

1. a discussion of the drawbacks of virtual communities of practice vis-a-vis face-to-face ones is made

2. "types of study" (p. 5) is further explained. Do you mean research methodology for reporting inquiries about CoP?

3. "Method of recruitment"  should include selection criteria (how it was decided who to contact and why), as well as argumentation made for persuading someone to participate.

4. "Theoretical frameworks ...." should be further explained. Do you mean frameworks that acted as motivators to form vCoP, theoretical frameworks used in the presentation and analysis of the vCoP?

5. a "Discussion" section is added.  This will give "depth" to the results obtained.

Author Response

Reviewer comment

Response

Line

1. a discussion of the drawbacks of virtual communities of practice vis-a-vis face-to-face ones is made.

Thank you. We feel this is an important topic and has been discussed at length in the introduction in our protocol paper (doi:10.1136/

bmjopen-2020-046998). I have now added some detail at the end of the first paragraph in the introduction.

54 - 58

2. "types of study" (p. 5) is further explained. Do you mean research methodology for reporting inquiries about CoP?

Thank you. The title has now been changed to ‘Research methodology’.

171

3. "Method of recruitment" should include selection criteria (how it was decided who to contact and why), as well as argumentation made for persuading someone to participate.

Section 3.8 details who was included in the VCoPs and section 3.9 outlines the roles within the VCoPs. Further detail on selection criteria is not included in the papers and is very specific to each VCoP. VCoP members are ‘invited’ to participate, and membership is voluntary, based on personal interest and desire to increase knowledge in a particular area.

233 - 267

4. "Theoretical frameworks ...." should be further explained. Do you mean frameworks that acted as motivators to form VCoP, theoretical frameworks used in the presentation and analysis of the VCoP?

I’ve added further detail to the title, it now reads ‘Theoretical frameworks supporting development of the VCoPs’. This now also matches the title of Table 5.

324

5. a "Discussion" section is added.  This will give "depth" to the results obtained.

The paper contains discussion, limitations and conclusions sections that discuss the results obtained (see sections 4 and 5). Some further detail has been added to the conclusions.

360 - 446

Round 2

Reviewer 1 Report

Thank you to the authors for addressing the reviewers' comments thoroughly. I have no objection to the manuscript's publication in its revised form.

Reviewer 3 Report

I am satisfied by the authors' response.